# Language-Aware Information Maximization for Transductive Few-Shot CLIP

**Ghassen Baklouti**                                                    *ghassen.baklouti.1@ens.etsmtl.ca*
*École de Technologie Supérieure (ÉTS), Montréal, Canada*

**Maxime Zanella**                                                      *maxime.zanella@uclouvain.be*
*Université Catholique de Louvain (UCLouvain), Louvain-La-Neuve, Belgium*
*Université de Mons (UMons), Mons, Belgium*

**Ismail Ben Ayed**                                                     *ismail.benayed@etsmtl.ca*
*École de Technologie Supérieure (ÉTS), Montréal, Canada*

## Abstract

Transductive few-shot learning has triggered an abundant literature focusing on vision-only models, but is still at a nascent stage within the recent context of foundational vision-language models (VLMs). Only a few recent methods addressed the problem, pointing to the potential of tranduction in VLMs and to the need for VLM-tailored methods. Building on this momentum, we leverage information-theoretic concepts and recent progress in parameter-efficient fine-tuning (PEFT), developing a highly competitive transductive few-shot CLIP method. Specifically, we introduce a novel Language-aware Information MaximizatiOn (LIMO) loss integrating three complementary terms: (i) the mutual information between the vision inputs and the textual class descriptions; (ii) a Kullback-Leibler (KL) divergence penalizing deviation of the network's probabilistic outputs from the text-driven zero-shot predictions; and (iii) a standard cross-entropy loss based on the labeled shots. Furthermore, we challenge the commonly followed fine-tuning practices in the context of transductive few-shot learning, and explore PEFT strategies, completely overlooked in this context. Surprisingly, we observe substantial boosts in performances, which points to the potential of adapting a subset of the model's parameters in the transductive few-shot setting. We report comprehensive evaluations, which show that LIMO outperforms the very recent transductive few-shot CLIP methods by a large margin and yields significant gains over the best-performing inductive methods. Our code is publicly available at: https://github.com/ghassenbaklouti/LIMO.

## 1 Introduction

The recent rapid advancements in hardware capabilities have driven substantial interest in multi-modal learning within the field of computer vision. In particular, vision-language models (VLMs) have received notable attention, evidenced by a sharp rise in related publications in recent years (Zhang et al., 2024). Leading this wave, CLIP (Radford et al., 2021) is a pioneering VLM that leverages natural language supervision, learning visual representations from a vast, task-agnostic dataset of 400 million image-text pairs. The pre-training process of CLIP involves jointly optimizing both image and text encoders in a contrastive fashion. This approach aligns paired images and text prompts in a shared embedding space, forcing the matched textual and visual representations to be close, while pushing non-matched pairs apart. Consequently, CLIP can perform robust zero-shot classification at the inference stage, where embeddings of test images are matched to the text-based prompts derived from the class names of the downstream task (e.g., `"a photo of a [class name]."`).

VLMs have demonstrated strong zero-shot performances across a wide range of tasks, yet they could be seriously challenged when dealing with fine-grained tasks that are significantly different from the pre-training

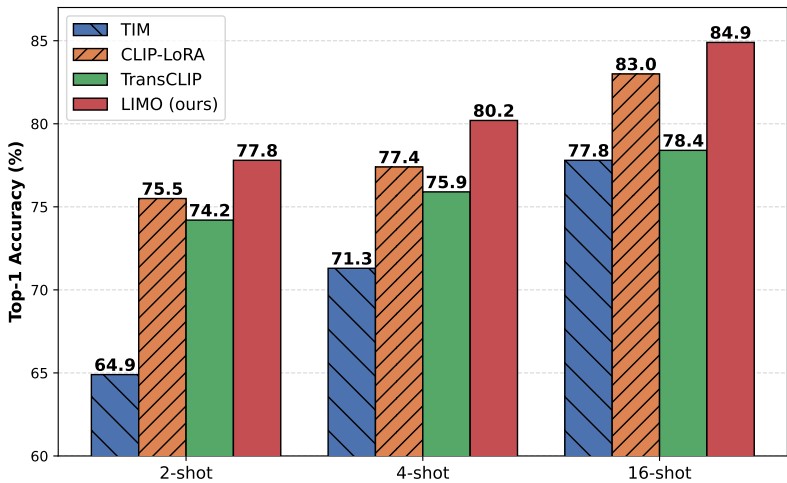

Figure 1: The reported performance is the average accuracy over the 11 datasets studied in this paper. LIMO outperforms a recent transductive VLM method (TransCLIP (Zanella et al., 2024)) and a related vision-only method (TIM (Boudiaf et al., 2020b)) by large margins. It also demonstrates a significant gain when compared to a related inductive method (CLIP-LoRA (Zanella & Ben Ayed, 2024a)).

data (Radford et al., 2021). To address this, ongoing research is currently being focused on adaptation methods, aiming at enhancing the generalization of VLMs and bridging the gap toward reliable deployments in real-world downstream applications (Zhou et al., 2022b;a; Zhang et al., 2022; Gao et al., 2024). Predominantly, these methods operate within an *inductive* framework, conducting inference independently for each test sample and omitting the relationships between the testing samples and their statistics. While effective, this approach may limit the model to isolated predictions, potentially overlooking valuable patterns within the overall target data distribution. The *transductive* setting is a powerful alternative, in which inference is performed jointly across the entire test dataset. This framework enables the model to leverage the statistical properties of the unlabeled target data as a whole, facilitating the discovery of shared structures as well as latent patterns that are missed when samples are processed independently. This paradigm has already demonstrated significant success in the context of vision-only models, with a large body of works that addressed transductive few-shot inference, e.g. (Liu et al., 2018; Dhillon et al., 2019; Ziko et al., 2020; Boudiaf et al., 2020b; Liu et al., 2020; Hu et al., 2021b), to cite a few. Currently, the transductive setting is gaining traction in VLMs, with several very recent works emerging on the subject (Martin et al., 2024; Kalantidis et al., 2024; Zanella et al., 2024; El Khoury et al., 2025).

These recent transductive few-shot methods for VLMs leverage the text-encoder knowledge, as a form of supervision, in conjunction with an unsupervised learning term, taking the form of a generative mixture model-based clustering (Martin et al., 2024; Zanella et al., 2024). One of the main driving motivations behind these recent VLM-tailored developments is that the standard transductive few-shot inference methods perform poorly with VLMs (Martin et al., 2024; Zanella et al., 2024), hence the need of dedicated techniques. As for the question "what model parameters to fine-tune", all the very recent transductive few-shot CLIP methods to our knowledge, i.e., (Martin et al., 2024; Zanella et al., 2024), focused on operating on the output embedding space, without exploring the possibility of fine-tuning the internal-representation parameters. This is, indeed, in line with the *de facto* choice in the abundant vision-only few-shot literature (Chen et al., 2019; Boudiaf et al., 2020b), in which fine-tuning focuses on the output layer, while freezing the remaining network-encoder parameters. In fact, full fine-tuning, i.e., fine-tuning the whole network is widely avoided in low-shot regimes as it is prone to over-fitting, substantially degrading the performances (Chen et al., 2019; Boudiaf et al., 2020b). However, in parallel, inductive few-shot learning methods for VLMs have shown impressive performance gains, building on the recent advances in Parameter-Efficient-Fine-Tuning (PEFT) strategies, which originated from and were popularized in NLP. This includes the seminal work of CoOp, which pioneered prompt learning in VLMs (Zhou et al., 2022b), as well as adapters (Zhang et al., 2022)

and, more recently, Low-Rank Adaptation (LoRA) (Zanella & Ben Ayed, 2024a). These recent inductive few-shot CLIP developments challenged this status quo in fine-tuning with low supervision, showing that updating the inner representations, or the input text prompts, could be beneficial even with a few labeled samples. Building on this momentum, we leverage information-theoretic concepts, together with these advances in PEFT strategies to push further the potential of the transductive paradigm in the context of VLMs. Specifically, our contributions could be summarized as follows:

**Main contributions.**

- We propose a novel transductive language-aware information-maximization loss. Our objective function aims at maximizing the mutual information between the vision inputs and the textual class descriptions, while penalizing deviation of the network's softmax outputs from the text-driven zero-shot predictions. It could be viewed as a generalization of information-maximization losses widely deployed in vision-only tasks.

- We challenge the commonly followed fine-tuning practices in the context of transductive few-shot learning, and explore recent advances in PEFT strategies, completely overlooked in this context. Surprisingly, we observe substantial boosts in performance, which points to the great potential of adapting a subset of the model parameters in the transductive few-shot setting.

- In Figure 1, our method, named **LIMO** (**L**anguage-aware **I**nformation **M**aximizati**O**n), outperforms a recent transductive method for VLMs by a large margin, especially in the higher-shot regimes, and also demonstrates significant gains over one of the best-performing inductive few-shot CLIP methods in the literature.

## 2 Related Work

**Transductive Inference.** In recent years, transductive inference has emerged as a highly effective approach in few-shot learning, where the objective is to generalize to new tasks or classes with very limited data. In contrast to traditional inductive methods, which solely rely on labeled training data (support set), transductive approaches utilize both labeled support samples and unlabeled test data (query samples). This enables to adjust predictions in response to the test set distribution, enhancing adaptability to new tasks. A seminal work that introduced transduction in vision-based few-shot learning proposed to use a meta-learned graph to propagate labels from the labeled support set to the unlabeled query set (Liu et al., 2018). Building on this, a range of other methods have since been developed. For instance, LaplacianShot (Ziko et al., 2020) exploits data structure via graph clustering, encouraging consistent class predictions among samples with similar features, while aligning each query sample with the closet support prototype; Transductive Fine-Tuning (TF) (Dhillon et al., 2019) incorporates entropy minimization on query samples as a regularization mechanism; TIM (Boudiaf et al., 2020b) maximizes the mutual information between the query images and the labels in conjunction with a cross-entropy on the support samples; BD-CSPN (Liu et al., 2020) refines class prototypes by addressing feature biases between the support and query sets, selectively updating prototypes based on the most confident query samples; and PT-MAP (Hu et al., 2021b) formulates an optimal transport problem to align the distributions of the support and query samples.

Despite their large potential, previously discussed transductive methods have been demonstrated to experience considerable performance declines when applied to VLMs (Zanella et al., 2024; Martin et al., 2024), as they rely exclusively on the visual features. This has prompted the development of several recent transductive approaches, designed to explicitly leverage the textual modality alongside image embeddings, a feature that was made possible by the advent of VLMs (Martin et al., 2024; Kalantidis et al., 2024; Zanella et al., 2024). The Dirichlet-based method in (Martin et al., 2024) used a maximum likelihood estimation objective function, while penalizing the total number of predicted classes in a given batch. TransCLIP (Zanella et al., 2024) also pursued a maximum likelihood formulation, but with different statistical models, and added a Kullback-Leibler divergence term penalizing deviation of the network's probability outputs from the zero-shot predictions. The ZLaP method in (Kalantidis et al., 2024) propagates the zero-shot labels based on the similarities between pairs of instances, following classical label-propagation principles.

**Parameter-Efficient Fine-Tuning.** Full fine-tuning (FFT) is one of the go-to-choice methods for adapting machine learning models to downstream tasks due to its excellent performance. This approach involves updating all the trainable parameters of a model during the adaptation process. However, for large-scale pre-trained models (often consisting of millions or even billions of parameters), FFT becomes less desirable as it requires substantial computational resources, leading to significant costs in terms of memory, time, and data requirements. To address this challenge, an alternative adaptation paradigm known as parameter-efficient fine-tuning (PEFT) has emerged. PEFT techniques aim to reduce the high expense associated with FFT by adjusting only a small subset of the model's parameters. Based on the choice of learnable parameters, PEFT methods can be grouped into four main categories: **Addition-based methods**, which introduce new trainable layers or adapters within the original frozen architecture to absorb task-specific knowledge (Zhang et al., 2022; Chen et al., 2022; Lian et al., 2022; Gao et al., 2024); **Selective-based methods**, which focus on updating a carefully-chosen subset of the model's existing parameters such as adjusting only the bias terms within certain blocks (Zaken et al., 2021) or fine tuning particular layers, like the last visual projector layer in CLIP (Fahes et al., 2024); **Prompt-tuning methods**, which involve adding tunable tokens to the input space or within intermediate sequences, enabling the model's outputs to be tailored for new tasks without altering its core parameters (Zhou et al., 2022b;a; Jia et al., 2022; Shu et al., 2022; Khattak et al., 2023; Ma et al., 2024); **Low-rank adaptation methods**, which use additional low rank matrices that are adaptable to approximate the change of weights during adaptation while keeping the original matrices frozen. LoRA (Hu et al., 2021a) pioneered this approach in the field of Natural Language Processing (NLP), and ever since, it has been adopted in vision tasks (Zanella & Ben Ayed, 2024a). Following LoRA, numerous variations aiming to make the rank adaptive (Valipour et al., 2022; Zhang et al., 2023), improve performance (Chavan et al., 2023; Kim et al., 2024), or enhance efficiency (Rajabzadeh et al., 2024; Dettmers et al., 2024), have emerged.

**Zero- and Few-Shot CLIP Adaptation.** With its extensive training on large-scale image-text data, CLIP has shown impressive performance across a variety of downstream tasks in zero-shot scenarios. However, for more specific tasks, it may not achieve the desired results (Radford et al., 2021). To tackle this issue, recent efforts have been directed towards exploring its full potential and further adapting it. Among these, prompt-tuning methods under limited supervision have become particularly prominent techniques (Zhou et al., 2022b;a; Zhu et al., 2023; Yao et al., 2023; Shu et al., 2022; Ma et al., 2024). CoOp (Zhou et al., 2022b) introduces a context optimization approach in which input text-prompt tokens are represented as continuous learnable vectors that are end-to-end optimized from the data. Building upon this, CoCoOp (Zhou et al., 2022a) integrates a meta-learning network to learn image-conditioned tokens, thereby improving model generalization to unseen classes. Other approaches, such as ProGrad (Zhu et al., 2023) and KgCoOp (Yao et al., 2023), focus on aligning learned prompts with predefined handcrafted ones. ProGrad (Zhu et al., 2023), for instance, uses gradient projection to guide prompt-tuning, aiming to preserve the model's original knowledge during the adaptation process. Additionally, prompt-tuning has been applied in test-time adaptation settings (Shu et al., 2022; Ma et al., 2024), wherein image-specific prompts are optimized in real time to maximize consistency across predictions generated from multiple augmentations of the same image.

Despite their effectiveness, prompt tuning methods remain computationally intensive, as they require multiple backpropagation passes through the entire text encoder. To mitigate these demands, several alternative approaches have been developed. For instance, MTA (Zanella & Ben Ayed, 2024b) introduces a novel test-time adaptation technique that optimizes visual features, providing an alternative to traditional test-time prompt tuning. Moreover, CLIP-adapter (Gao et al., 2024) and Tip-adapter (Zhang et al., 2022) propose visual classifiers at the output of the vision encoder to combine adapted and original features. LP++ (Huang et al., 2024) extends the Linear-Probe baseline by incorporating knowledge from the text encoder. ProLIP (Fahes et al., 2024) limits updates to CLIP's last visual projector layer while CLIP-LoRA (Zanella & Ben Ayed, 2024a) leverages low-rank matrices (LoRA) to adapt the attention matrices of both vision and text encoders at the same time.

**Information Maximization.** Information Maximization has been widely used in machine learning and computer vision tasks including representation learning (Tschannen et al., 2019; Hjelm et al., 2018; Bachman et al., 2019; Kemertas et al., 2020), deep clustering (Hu et al., 2017; Jabi et al., 2019; Krause et al., 2010),

metric learning (Boudiaf et al., 2020a), domain adaptation (Pan et al., 2020), semi-supervised learning (Chiaroni et al., 2023) and few-shot learning (Boudiaf et al., 2020b). This approach aims to maximize the mutual information between two sets of variables, leading to more informative and discriminative representations. While extensively studied for vision-only classifiers, this strategy, to the best of our knowledge, has not yet been applied to VLMs.

## 3   Method

In this section, we first introduce some basic notation and provide a detailed presentation of the proposed transductive Language-aware Information MaximizatiOn (LIMO) loss and the fine-tuning strategy to optimize it.

### 3.1   CLIP Few-Shot Adaptation

Assume that we are given $K$ candidate classes for a zero-shot classification task using a VLM, such as CLIP (Radford et al., 2021). In such a setting, textual descriptions, often referred to as "prompts", are created for each class, e.g. $\mathbf{c}_k = $ `"a photo of a [kth class name]."`, $k = 1, ..., K$. Each prompt is then embedded into a normalized representation on the unit hypersphere, $\mathbf{t}_k = f(\mathbf{c}_k, \boldsymbol{\theta}_t)$, where $f$ is the language encoder and $\boldsymbol{\theta}_t$ its vector of learnable parameters. For a given image classification dataset (or task), we have $N$ images, $\mathbf{x}_i$, $i = 1, ..., N$, including both labeled and unlabeled ones. Let $\mathbb{S} \subset \{1, ..., N\}$ denotes the set of sample indices within the set of labeled images in the data, often called *the support set* in the few-shot literature. Also, $\mathbb{Q} \subset \{1, ..., N\}$ contains the indices of the unlabeled samples (for which we need to make predictions), called *the query set*.

Each image $\mathbf{x}_i$ in the data, $i = 1, ..., N$, is mapped to a normalized embedding space having the same dimensionality as the text-embedding space, using vision encoder $g$ parameterized by $\boldsymbol{\theta}_v$. This yields, for each image $\mathbf{x}_i$, a visual embedding $\mathbf{f}_i = g(\mathbf{x}_i, \boldsymbol{\theta}_v)$. For zero-shot prediction, a straightforward approach for applying VLMs to downstream classification tasks, each text embedding $\mathbf{t}_k$ is paired with the visual embedding $\mathbf{f}_i$ of a test image $\mathbf{x}_i$, and their cosine similarity is computed to produce a prediction logit:

$$l_{i,k} = \mathbf{f}_i^\top \mathbf{t}_k \tag{1}$$

This logit is then converted into a prediction in the form of a posterior softmax probability of the class' textual description, $\mathbf{c}_k$, given test input $\mathbf{x}_i$:

$$p_{i,k} = \frac{\exp(l_{i,k}/\tau)}{\sum_j^K \exp(l_{i,j}/\tau)} \tag{2}$$

where $\tau$ is a temperature scaling parameter. Then, the zero-shot classification of an image $\mathbf{x}_i$ is performed by selecting the class with the highest posterior probability: $\hat{k} = \arg\max_k p_{i,k}$.

In the standard few-shot setting, we assume that we have access to $|\mathbb{S}|/K$ labeled samples for each target class, with $|.|$ denoting the cardinality of a set. Typically, $|\mathbb{S}|/K$, the number of shots per class, is small (less than 16). Let $z_{ik}$ denote the one-hot encoded label for a labeled support image $\mathbf{x}_i$, with $z_{ik} = 1$ if $k$ is the class of image $\mathbf{x}_i$ and 0 otherwise. In the inductive setting (Zanella & Ben Ayed, 2024a), model adaptation is done by minimizing the cross-entropy (CE) loss with respect to the parameters of both the text and vision encoders:

$$C(\boldsymbol{\theta}_v, \boldsymbol{\theta}_t) = -\frac{1}{|\mathbb{S}|} \sum_{i \in \mathbb{S}} \sum_{k=1}^K z_{ik} \ln p_{i,k} \tag{3}$$

### 3.2   Language-aware information maximization

Inspired by previous works on vision-only transductive few-shot learning, such as TIM (Boudiaf et al., 2020b), we propose to minimize with respect to $\boldsymbol{\theta}_v$ and $\boldsymbol{\theta}_t$ a generalization of the mutual-information loss to VLMs:

$$L(\boldsymbol{\theta}_v, \boldsymbol{\theta}_t) = \underbrace{C(\boldsymbol{\theta}_v, \boldsymbol{\theta}_t)}_{\text{Cross Entropy}} - \underbrace{I(\boldsymbol{\theta}_v, \boldsymbol{\theta}_t)}_{\text{Mutual Info.}} + \underbrace{T(\boldsymbol{\theta}_v, \boldsymbol{\theta}_t)}_{\text{Text Reg.}} \tag{4}$$

In the following, we describe the notations occurring in our model in Eq. (4), as well as the effect of each of its terms:

**The mutual Information.** $I(\boldsymbol{\theta}_v, \boldsymbol{\theta}_t)$ is the mutual information between the visual inputs and the textual class representations, considering both the input images and the text prompts as random variables, denoted $\mathcal{X}$ and $\mathcal{C}$, respectively. We compute the posterior distribution over the textual class descriptions, given the input query images, using the zero-shot predictions:

$$\Pr(\mathcal{C} = \mathbf{c}_k | \mathcal{X} = \mathbf{x}_i; \boldsymbol{\theta}_v, \boldsymbol{\theta}_t) = p_{i,k} \tag{5}$$

and the marginal distribution of the textual descriptions over the query set as follows:

$$\Pr(\mathcal{C} = \mathbf{c}_k; \boldsymbol{\theta}_v, \boldsymbol{\theta}_t) = \frac{1}{|\mathbb{Q}|} \sum_{i \in \mathbb{Q}} p_{i,k} = \bar{p}_k \tag{6}$$

This part of the objective integrates two terms, conditional entropy $H(\mathcal{C}|\mathcal{X}; \boldsymbol{\theta}_v, \boldsymbol{\theta}_t)$ and marginal entropy $H(\mathcal{C}; \boldsymbol{\theta}_v, \boldsymbol{\theta}_t)$, as detailed in the following:

$$I(\boldsymbol{\theta}_v, \boldsymbol{\theta}_t) = \lambda_{ent} H(\mathcal{C}; \boldsymbol{\theta}_v, \boldsymbol{\theta}_t) - \lambda_{cond} H(\mathcal{C}|\mathcal{X}; \boldsymbol{\theta}_v, \boldsymbol{\theta}_t) \tag{7}$$

where $\lambda_{ent}$ and $\lambda_{cond}$ are two nonnegative weighting factors that control the influence of each term, $H(\mathcal{C}|\mathcal{X}; \boldsymbol{\theta}_v, \boldsymbol{\theta}_t)$ is the empirical estimate of the conditional entropy expressed as a function of the language-driven predictions as follows:

$$H(\mathcal{C}|\mathcal{X}; \boldsymbol{\theta}_v, \boldsymbol{\theta}_t) \propto - \sum_{k=1}^{K} \sum_{i \in \mathbb{Q}} \Pr(\mathbf{c}_k | \mathcal{X} = \mathbf{x}_i; \boldsymbol{\theta}_v, \boldsymbol{\theta}_t) \ln \Pr(\mathbf{c}_k | \mathcal{X} = \mathbf{x}_i; \boldsymbol{\theta}_v, \boldsymbol{\theta}_t) = - \sum_{k=1}^{K} \sum_{i \in \mathbb{Q}} p_{i,k} \ln p_{i,k} \tag{8}$$

and $H(\mathcal{C}; \boldsymbol{\theta}_v, \boldsymbol{\theta}_t)$ denotes the empirical estimate of the marginal entropy given by:

$$H(\mathcal{C}; \boldsymbol{\theta}_v, \boldsymbol{\theta}_t) = - \sum_{k=1}^{K} \Pr(\mathcal{C} = \mathbf{c}_k; \boldsymbol{\theta}_v, \boldsymbol{\theta}_t) \ln \Pr(\mathcal{C} = \mathbf{c}_k; \boldsymbol{\theta}_v, \boldsymbol{\theta}_t) = - \sum_{k=1}^{K} \bar{p}_k \ln \bar{p}_k \tag{9}$$

Each entropy term plays a distinct and complementary role during adaptation. On the one hand, $H(\mathcal{C}|\mathcal{X}; \boldsymbol{\theta}_v, \boldsymbol{\theta}_t)$ aims to minimize the uncertainty associated with the text-based prediction of each query sample. On the other hand, $H(\mathcal{C}; \boldsymbol{\theta}_v, \boldsymbol{\theta}_t)$ promotes a balanced distribution between classes, encouraging diversity in class predictions, and preventing the model from favoring a limited number of classes. Thus, integrating the marginal entropy serves as a regularization term preventing degenerate solutions that may arise when solely minimizing conditional entropy. Together, these entropy terms enable the model to produce confident and well-distributed predictions on the unlabeled samples.

**The text-based regularization.** This term provides text-based supervision, as it minimizes a Kullback-Leibler (KL) divergence between the network's probability outputs, $\mathbf{p}_i = (p_{i,k})_{1 \le k \le K}$, and the zero-shot predictions (i.e., the initial network's outputs), which we denote $\hat{\mathbf{y}}_i = (y_{i,k})_{1 \le k \le K}$:

$$T(\boldsymbol{\theta}_t, \boldsymbol{\theta}_v) = \lambda_{text} \sum_{i \in \mathbb{Q}} \mathrm{KL}(\mathbf{p}_i || \hat{\mathbf{y}}_i) \tag{10}$$

with

$$\mathrm{KL}(\mathbf{p}_i || \hat{\mathbf{y}}_i) = \mathbf{p}_i^{\top} \log \frac{\mathbf{p}_i}{\hat{\mathbf{y}}_i}, \quad i \in \mathbb{Q}. \tag{11}$$

and $\lambda_{text}$ is a nonnegative weighting factor enabling to control the contribution of this term to our overall loss. This text-based regularizer helps preserve the zero-shot generalization capabilities of CLIP, penalizing large deviations of predictions $\mathbf{p}_i$ from initial zero-shot predictions $\hat{\mathbf{y}}_i$.

### 3.3 Umbrella method

Thanks to its structure, which simultaneously incorporates both visual and textual feature embeddings, and both labeled and unlabeled data, our loss could be viewed as a generalization of state-of-the-art methods.

**Generalization of TIM (Boudiaf et al., 2020b).**  Since TIM is a transductive information maximization approach initially designed for vision-only classifiers, it relies only on vision features and omits textual elements. By setting $\lambda_{text} = 0$ in our loss function $L(\boldsymbol{\theta}_v, \boldsymbol{\theta}_t)$ in equation 4, and fixing the text encoder to act just as a linear classifier, whose weights are given by the text embedding, we can perceive TIM as a specific case of LIMO.

**Generalization of CLIP-LoRA (Zanella & Ben Ayed, 2024a).**  CLIP-LoRA leverages low-rank matrices to adapt the vision and test encoders of CLIP. This method operates in a purely inductive manner that relies exclusively on labeled data without incorporating unlabeled data. By setting $\lambda_{text} = 0$, $\lambda_{ent} = 0$ and $\lambda_{cond} = 0$ in our loss function, we recover CLIP-LoRA as a specific instance of LIMO.

### 3.4 Low-Rank Adaptation

In the following, we present our fine-tuning strategy based on LoRA (Hu et al., 2021a) to adapt vision and text encoders. This choice is motivated by the fact that full fine-tuning is well known to produce degenerate solutions (Chen et al., 2019; Boudiaf et al., 2020b), hence a more efficient way to apply LIMO is required. LoRA approximates incremental updates of the pre-trained weights as the product of two small matrices, $\mathbf{A}$ and $\mathbf{B}$, based on the concept of ''intrinsic rank'' of the downstream task where $\mathbf{A} \in \mathrm{R}^{r \times n}$, $\mathbf{B} \in \mathrm{R}^{m \times r}$ are two low-rank matrices and $r$ typically $<< m, n$. This low-rank update is then incorporated into the forward pass of a specific network layer $l$. The output hidden state $\mathbf{h}^{(l)}$ is computed using the hidden state from the preceding layer, $\mathbf{h}^{(l-1)}$, and the layer's original, frozen weight matrix, $\mathbf{W}^{(l)}$, as follows:

$$\mathbf{h}^{(l)} = \mathbf{W}^{(l)}\mathbf{h}^{(l-1)} + \Delta\mathbf{W}^{(l)}\mathbf{h}^{(l-1)} = \mathbf{W}^{(l)}\mathbf{h}^{(l-1)} + \gamma\mathbf{B}^{(l)}\mathbf{A}^{(l)}\mathbf{h}^{(l-1)} \tag{12}$$

where $\gamma$ is a scaling factor that controls the influence of the low-rank updates. Values in matrix $\mathbf{A}^{(l)}$ are randomly initialized via Kaiming initialization and $\mathbf{B}^{(l)}$ is filled with zeros so that there is no incremental update before training.

## 4 Experiments

### 4.1 Datasets.

We evaluate our method on eleven publicly available image classification datasets, which have already been used in prior studies (Zhou et al., 2022b; Zanella et al., 2024; Martin et al., 2024). These datasets include ImageNet (Deng et al., 2009), SUN (Xiao et al., 2010), FGVC (Maji et al., 2013), EuroSAT (Helber et al., 2019), Cars(Krause et al., 2013), Food(Bossard et al., 2014), Pets(Parkhi et al., 2012), Flowers (Nilsback & Zisserman, 2008), Caltech (Fei-Fei et al., 2004), DTD (Cimpoi et al., 2014) and UCF (Soomro, 2012). Unless otherwise specified, all experiments were conducted with the pre-trained ViT-B/16 vision encoder from CLIP, and the numerical results are presented as top-1 accuracy, averaged over three distinct random seeds.

### 4.2 Implementation details

**Our fine-tuning strategy.**  We follow the design choices of (Zanella & Ben Ayed, 2024a) by applying LoRA to all the query, key, and value matrices of the vision and text encoders of CLIP, using a rank of $r = 2$. A dropout layer with $p = 0.25$ is used to regularize the input to the LoRA module. In the few-shot setting, the number of iterations is set to 500 times $|\mathbb{S}|/K$ (the number of labeled samples per class). Unless otherwise specified, these hyperparameters remain fixed across all experiments.

**LIMO hyperparameters.** The core component of our transductive formulation is the language-aware information maximization loss function, which comprises three terms, see Eq. (4). We set the conditional entropy weight $\lambda_{cond}$ to 1, the marginal entropy weight $\lambda_{ent}$ to 10, and the text regularization weight $\lambda_{text}$ to 0.1. To ensure simplicity and generalizability throughout the experiments, these hyperparameters are kept constant across all experiments.

## 4.3 Results

**LIMO overall performances.** Table 1 highlights our results in comparison with state-of-the-art few-shot learning methods. We can observe that LIMO consistently achieves leading performances across all datasets in the 4-shot and 16-shot settings and across 10 datasets in the 2-shot setting, yielding significant improvements over both transductive and inductive methods. For instance, LIMO outperforms TransCLIP (Zanella et al., 2024) by an average of 5% in the 4-shot setting.

Table 1: **Comparison of state-of-the-art methods in few-shot learning**: Top-1 classification accuracy on 11 datasets. **Bolded** values indicate highest accuracy. ✓ denotes transductive methods, ✗ denotes non-transductive methods.

| | | | ImageNet | Sun | FGVC | EuroSAT | Cars | Food | Pets | Flowers | Caltech | DTD | UCF | Average |
|---|---|---|---|---|---|---|---|---|---|---|---|---|---|---|
| | | Zero-Shot | 66.7 | 62.6 | 24.7 | 47.5 | 65.3 | 86.1 | 89.1 | 71.4 | 92.9 | 43.6 | 66.7 | 65.1 |
| 2-shot | Vision-only | TF (Dhillon et al., 2019) ✓ | 40.5 | 51.6 | 25.3 | 63.1 | 45.1 | 58.8 | 54.8 | 83.2 | 87.0 | 47.3 | 59.4 | 56.0 |
| | | BD-CSPN (Liu et al., 2020) ✓ | 46.1 | 56.1 | 26.7 | 64.7 | 50.7 | 67.5 | 64.6 | 89.6 | 89.6 | 48.9 | 64.0 | 60.8 |
| | | LaplacianShot (Ziko et al., 2020) ✓ | 45.8 | 55.9 | 27.1 | 68.2 | 51.1 | 68.2 | 66.0 | 89.7 | 89.6 | 48.9 | 65.1 | 61.4 |
| | | PT-MAP (Hu et al., 2021b) ✓ | 50.7 | 63.1 | 28.6 | 71.7 | 57.5 | 77.5 | 75.7 | 73.9 | 59.1 | 53.8 | 68.7 | 61.9 |
| | | TIM (Boudiaf et al., 2020b) ✓ | 47.9 | 60.7 | 28.1 | 75.8 | 55.7 | 78.7 | 70.6 | 91.4 | 86.6 | 52.3 | 66.4 | 64.9 |
| | Vision-Language | CoOp (Zhou et al., 2022b) ✗ | 67.0 | 67.0 | 25.9 | 65.1 | 70.4 | 84.4 | 89.9 | 88.0 | 93.1 | 54.1 | 74.1 | 70.8 |
| | | Tip-Adapter-F (Zhang et al., 2022) ✗ | 70.0 | 68.6 | 32.8 | 73.2 | 70.8 | 86.0 | 91.6 | 90.1 | 93.9 | 57.8 | 76.2 | 73.7 |
| | | LP++ (Huang et al., 2024) ✗ | 69.65 | 70.12 | 32.27 | 67.81 | 70.46 | 86.14 | 90.19 | 90.39 | 94.14 | 56.34 | 74.85 | 72.94 |
| | | CLIP-LoRA (Zanella & Ben Ayed, 2024a) ✗ | 70.8 | 71.3 | 32.7 | 82.7 | 73.2 | 83.2 | 91.3 | 89.8 | 94.6 | 59.9 | 80.0 | 75.5 |
| | | TransCLIP (Zanella et al., 2024) ✓ | 70.3 | 70.9 | 30.0 | 77.1 | 71.7 | **87.0** | 91.7 | 90.6 | 93.5 | 55.1 | 78.5 | 74.2 |
| | | LIMO (Ours) ✓ | **71.1** | **72.3** | **33.8** | **91.7** | **74.5** | 86.3 | **93.8** | **92.4** | **94.9** | **63.3** | **81.2** | **77.8** |
| 4-shot | Vision-only | TF ✓ | 51.1 | 61.0 | 30.3 | 64.9 | 56.8 | 71.0 | 65.9 | 90.9 | 91.5 | 53.7 | 67.9 | 64.1 |
| | | BD-CSPN ✓ | 53.8 | 62.5 | 30.5 | 64.8 | 58.5 | 75.3 | 72.0 | 92.5 | 92.0 | 52.1 | 70.9 | 65.9 |
| | | LaplacianShot ✓ | 53.5 | 62.5 | 29.6 | 74.3 | 58.5 | 75.7 | 73.4 | 92.8 | 92.0 | 52.7 | 71.7 | 67.0 |
| | | PT-MAP ✓ | 57.6 | 68.1 | 31.2 | 74.9 | 63.1 | 81.1 | 79.5 | 76.2 | 60.2 | 58.4 | 73.9 | 65.8 |
| | | TIM ✓ | 57.4 | 67.0 | 32.8 | 79.3 | 65.8 | 83.5 | 82.3 | 93.4 | 88.5 | 58.1 | 76.5 | 71.3 |
| | Vision-Language | CoOp ✗ | 68.8 | 69.7 | 30.9 | 69.7 | 74.4 | 84.5 | 92.5 | 92.2 | 94.5 | 59.5 | 77.6 | 74.0 |
| | | Tip-Adapter-F ✗ | 70.7 | 70.8 | 35.7 | 76.8 | 74.1 | 86.5 | 91.9 | 92.1 | 94.8 | 59.8 | 78.1 | 75.6 |
| | | LP++ ✗ | 70.79 | 73.21 | 34.39 | 74 | 74.53 | 85.96 | 91.08 | 92.92 | 95.06 | 62.27 | 79.55 | 75.8 |
| | | CLIP-LoRA ✗ | 71.4 | 72.8 | 37.9 | 84.9 | 77.4 | 82.7 | 91.0 | 93.7 | 95.2 | 63.8 | 81.1 | 77.4 |
| | | TransCLIP ✓ | 70.3 | 71.9 | 34.0 | 79.4 | 74.0 | 86.4 | 91.6 | 93.6 | 94.0 | 61.1 | 79.1 | 75.9 |
| | | LIMO (Ours) ✓ | **71.9** | **74.1** | **40.3** | **93.1** | **78.6** | **86.6** | **94.3** | **96** | **95.8** | **67.6** | **84** | **80.2** |
| 16-shot | Vision-only | TF ✓ | 61.8 | 70.1 | 38.3 | 74.3 | 71.2 | 80.7 | 79.5 | 95.4 | 93.6 | 62.9 | 76.0 | 73.1 |
| | | BD-CSPN ✓ | 61.7 | 69.4 | 37.7 | 73.4 | 70.7 | 80.2 | 81.2 | 94.8 | 93.3 | 61.3 | 76.0 | 72.7 |
| | | LaplacianShot ✓ | 60.9 | 68.3 | 36.1 | 78.1 | 69.2 | 81.2 | 81.7 | 94.8 | 93.1 | 58.6 | 76.3 | 72.6 |
| | | PT-MAP ✓ | 64.0 | 72.0 | 37.4 | 75.6 | 72.0 | 82.7 | 86.1 | 78.5 | 63.7 | 63.7 | 76.3 | 70.2 |
| | | TIM ✓ | 67.8 | 73.6 | 40.6 | 83.6 | 79.5 | 84.9 | 88.7 | 95.4 | 92.4 | 67.5 | 82.1 | 77.8 |
| | Vision-Language | CoOp ✗ | 71.9 | 74.9 | 43.2 | 85.0 | 82.9 | 84.2 | 92.0 | 96.8 | 95.8 | 69.7 | 83.1 | 80.0 |
| | | Tip-Adapter-F ✗ | 73.4 | 76.0 | 44.6 | 85.9 | 82.3 | 86.8 | 92.6 | 96.2 | 95.7 | 70.8 | 83.9 | 80.7 |
| | | LP++ ✗ | 72.96 | 76.07 | 41.46 | 84.68 | 80.64 | 87.29 | 92.88 | 96.45 | 95.89 | 71.39 | 83.69 | 80.31 |
| | | CLIP-LoRA ✗ | 73.6 | 76.1 | 54.7 | 92.1 | 86.3 | 84.2 | 92.4 | 98.0 | 96.4 | 72.0 | 86.7 | 83.0 |
| | | TransCLIP ✓ | 71.8 | 74.7 | 38.6 | 83.0 | 79.8 | 86.9 | 92.4 | 94.4 | 94.0 | 65.1 | 82.1 | 78.4 |
| | | LIMO (Ours) ✓ | **74** | **77.7** | **57.5** | **94.4** | **87.4** | **87.4** | **95.3** | **98.7** | **97.1** | **74.7** | **89.3** | **84.9** |

**Comparison with vision-only classifiers.** Transductive few-shot learning methods, designed specifically for vision-only-classifiers, rely exclusively on visual information, omitting textual elements. By comparing LIMO to these approaches, we can assess the impact of incorporating the textual modality to enhance our mutual information regularization term. Table 1 demonstrates that integrating language within our transductive framework leads to substantial gains over all vision-only methods. Notably, our loss achieves

an average improvement exceeding 12% in the 2-shot setting compared to the TIM method (Boudiaf et al., 2020b) based only on vision features.

**Transduction improvements.**   We compare LIMO with state-of-the-art inductive few-shot methods, primarily developed for VLMs. This comparison enables us to evaluate the impact of our loss in effectively leveraging both textual and visual information in transductive fashion to enhance model performance. As shown in Table 1, our transductive approach significantly outperforms inductive VLMs across all tasks, particularly achieving an average improvement of 2.8% over CLIP-LoRA (Zanella & Ben Ayed, 2024a) in the 4-shot scenario. Interestingly, while other works have reported diminished transductive gains as the number of shots increases (Zanella et al., 2024)—attributing this to inductive methods better capturing the underlying data structure with more samples—LIMO consistently delivers significant improvements over inductive approaches, even in the 16-shot scenario. This highlights that transduction can be highly valuable across varying levels of supervision.

**LIMO's behavior.**   We further analyze how dataset properties influence LIMO's gains by comparing our results to CLIP's zero-shot performance. We observe that datasets exhibiting a substantial distributional shift from CLIP's pretraining distribution, such as Eurosat, DTD and FGVC, show the largest improvement compared to other datasets. This trend aligns with LIMO's objective, which maximizes the mutual information between visual inputs and textual class representations, particularly effective when the default alignment between CLIP's encoders is weak. In contrast, datasets on which the model already achieves high zero-shot accuracy (e.g., Food), tend to exhibit smaller gains in low-shot settings, and may even under-perform, as the limited guidance provided by the support set can cause LIMO to slightly over-regularize. This behavior is reflected in the 2-shot results for Food reported in Table 1.

**Statistical analysis.**   Figure 2 plots additional results showing error bars (conventional 95%confidence intervals) averaged across four representative datasets including fine-grained (FGVC), texture (DTD), satellite (Eurosat) and generic (Caltech) over 10 random seeds. This comparison includes the top competing methods: TransCLIP (Zanella et al., 2024), CLIP-LoRA (Zanella & Ben Ayed, 2024a), and our proposed LIMO under the 2-, 4- and 16-shot settings. One may observe that there are no overlapping error bars between LIMO and other methods, which suggests statistically significant improvements relative to the state of the art.

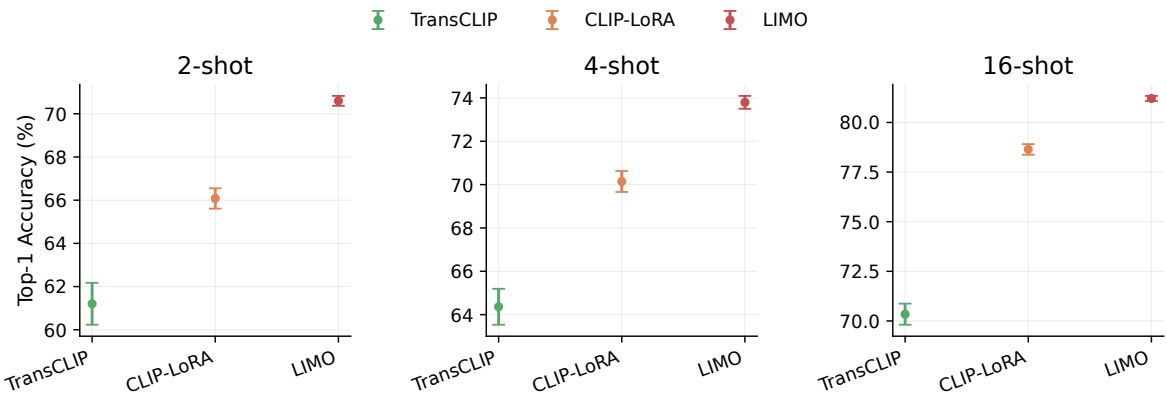

Figure 2: Statistical comparison of LIMO, TransCLIP (Zanella et al., 2024), and CLIP-LoRA (Zanella & Ben Ayed, 2024a) under the 2-, 4- and 16-shot settings. Error bars represent the standard deviation averaged across four datasets (FGVC, EuroSAT, Caltech, and DTD) using 10 seeds.

## 4.4   Ablation studies

**LIMO performances are consistent across various vision encoders.**   As presented in Table 2, LIMO outperforms leading transductive approaches for vision-language models (TransCLIP (Zanella et al., 2024))

and vision-only classifiers (TIM (Boudiaf et al., 2020b)) with both the ViT-B/32 architecture and the larger ViT-L/14. This performance advantage also holds when compared to one of the best-performing inductive few-shot methods in the literature, CLIP-LoRA (Zanella & Ben Ayed, 2024a), thereby highlighting the versatility of LIMO. Detailed results for both backbones are provided in the Appendix (Table 6 and 7).

Table 2: **Average Top-1 accuracy over 11 datasets for the ViT-B/32 and ViT-L/14 visual backbones**. **Bolded** values indicate highest accuracy. ✓ denotes transductive methods, ✗ denotes non-transductive methods.

|  |  |  | 2-shot | 4-shot | 16-shot |
|---|---|---|---|---|---|
|  | TIM (Boudiaf et al., 2020b) | ✓ | 58.7 | 65.5 | 73.3 |
|  | CoOp (Zhou et al., 2022b) | ✗ | 65.5 | 69.5 | 75.7 |
| ViT-B/32 | CLIP-LoRA (Zanella & Ben Ayed, 2024a) | ✗ | 70.7 | 72.9 | 78.9 |
|  | TransCLIP (Zanella et al., 2024) | ✓ | 70 | 71.5 | 74.3 |
|  | LIMO (ours) | ✓ | **73.2** | **75.7** | **81.1** |
|  | TIM | ✓ | 73.2 | 78 | 83.4 |
|  | CoOp | ✗ | 76.9 | 79.4 | 84.6 |
| ViT-L/14 | CLIP-LoRA | ✗ | 80.9 | 82.7 | 86.9 |
|  | TransCLIP | ✓ | 79.9 | 81.2 | 83.7 |
|  | LIMO (ours) | ✓ | **83.2** | **85.1** | **88.6** |

**Components of LIMO.** We assess the contribution of each term in our loss function on the overall performance of LIMO. The results, shown in Table 3, demonstrate that the full integration of all four components achieves better results than any other configuration. Interestingly, removing the mutual information term, $I(\boldsymbol{\theta}_v, \boldsymbol{\theta}_t)$, reduces significantly the performance of LIMO, particularly when only the last visual projector is updated. This outcome can be attributed to the mutual information term's role in improving the model's confidence over individual decisions while promoting a balanced distribution of predictions across classes. Additionally, integrating the text-based regularization term $T(\boldsymbol{\theta}_v, \boldsymbol{\theta}_t)$ with the mutual information increases the general performance of LIMO for both LoRA and LVP fine-tuning strategy. Aligning the model's predictions with the zero-shot predictions may help the model to preserve relevant zero-shot generalization properties and prevent overfitting. These results highlight the importance of each term in Eq. (4).

**Effects of different fine-tuning strategies.** While LIMO is optimized based on LoRA, alternative fine-tuning strategies exist such as adapting the textual input (e.g., CoOp (Zhou et al., 2022b)) or selectively adjust some specific components of the model (e.g., ProLIP (Fahes et al., 2024)). To assess different approaches, we introduce two LIMO variants in Table 3: LIMO-CoOp, in which the textual input is fine-tuned as a set of trainable vectors, and LIMO-LVP, where only the last visual projector layer is updated. Across nearly all configurations, LIMO (using LoRA) outperforms both variants by an average margin of 4%, suggesting that the parameters requiring adaptation for optimal performance are likely distributed throughout the model rather than concentrated solely in the final visual projection layer or the textual input. In one specific configuration (when only text-based regularization term $T(\boldsymbol{\theta}_v, \boldsymbol{\theta}_t)$ is integrated), LIMO-CoOp outperforms the LoRA-based variant by approximately 1%.

**Orthogonality of LIMO.** Table 3 also allows us to assess the compatibility of LIMO's objective function with other methods, such as CoOp (Zhou et al., 2022b) and ProLIP (Fahes et al., 2024). As reported, integrating our objective function leads to consistent performance gains—improving CoOp by approximately 1% and ProLIP by around 5%. These results highlight the orthogonality of LIMO and its effectiveness in enhancing the performance of other adaptation methods.

**Computational Requirements.** LIMO's cost varies based on several factors such as batch size and the scope of parameter adaptation (e.g., whether one or both encoders are fine-tuned). However, it remains minimal due to its orthogonality. For instance, integrating LIMO with CLIP-LoRA, as in the paper, increases training time by only $\sim 5$ minutes when adapting CLIP ViT-B/16 to ImageNet on a single NVIDIA RTX A6000 GPU.

**Sensitivity analysis of LIMO's hyperparameters.** Table 4 presents a comprehensive analysis of the impact of the regularization weights $\lambda_{ent}$, $\lambda_{cond}$, and $\lambda_{text}$ on model performance. We observe that increasing

Table 3: **Ablation study** on the contribution of each term of our proposed loss function LIMO under different fine-tuning strategies: LoRA (Low-Rank Adaptation), LVP (fine-tuning only the Last Visual Projector of CLIP), and CoOp (learning only the textual prompts). We report 4-shot results as top-1 accuracy and the average top-1 accuracy across 10 datasets (excluding ImageNet).

| Method | Loss | Sun | FGVC | EuroSAT | Cars | Food | Pets | Flowers | Caltech | DTD | UCF | Average |
|---|---|---|---|---|---|---|---|---|---|---|---|---|
| LoRA (Hu et al., 2021a) | $\mathcal{C}(\theta_v,\theta_t)$ | 72.78 | 37.69 | 84.76 | 77.41 | 82.96 | 90.39 | 94.19 | 95.39 | 64.19 | 80.77 | 78.05 |
| | $\mathcal{C}(\theta_v,\theta_t) - \mathcal{I}(\theta_v,\theta_t)$ | 73.69 | 39.72 | 91.83 | 78.11 | 85.89 | 93.98 | **96.78** | 95.73 | 68.3 | 83.87 | 80.79 |
| | $\mathcal{C}(\theta_v,\theta_t) + \mathcal{T}(\theta_v,\theta_t)$ | 73.09 | 36.51 | 56.65 | 77.82 | 84.77 | 90.14 | 91.77 | 95.11 | 54.87 | 79.68 | 74.04 |
| | $\mathcal{C}(\theta_v,\theta_t) - \mathcal{I}(\theta_v,\theta_t) + \mathcal{T}(\theta_v,\theta_t)$ | **73.94** | **39.78** | **92.69** | **78.75** | **86.67** | **94.3** | 96.08 | **95.91** | 67.93 | **84.01** | **81.01** |
| CoOp (Zhou et al., 2022b) | $\mathcal{C}(\theta_v,\theta_t)$ | 71.22 | 33.67 | 76.2 | 72.03 | 84.69 | 89.91 | 92.11 | 94.92 | 59.1 | 77.45 | 75.13 |
| | $\mathcal{C}(\theta_v,\theta_t) - \mathcal{I}(\theta_v,\theta_t)$ | 71.11 | 31.69 | **76.98** | 70.87 | **86.05** | **92.61** | 93.12 | 94.97 | 62.96 | **78.87** | 75.92 |
| | $\mathcal{C}(\theta_v,\theta_t) + \mathcal{T}(\theta_v,\theta_t)$ | **71.32** | **33.76** | 75.28 | **72.06** | 84.61 | 89.9 | 92.08 | 94.86 | 57.73 | 77.39 | 74.9 |
| | $\mathcal{C}(\theta_v,\theta_t) - \mathcal{I}(\theta_v,\theta_t) + \mathcal{T}(\theta_v,\theta_t)$ | 71.18 | 31.77 | 76.52 | 70.82 | 86.04 | 92.6 | **93.29** | **95.02** | **63.2** | 78.77 | 75.92 |
| LVP (Fahes et al., 2024) | $\mathcal{C}(\theta_v,\theta_t)$ | 69.29 | 34.42 | 75.96 | 72.44 | 77.68 | 82.38 | 93.86 | 93.47 | 62.18 | 75.65 | 73.73 |
| | $\mathcal{C}(\theta_v,\theta_t) - \mathcal{I}(\theta_v,\theta_t)$ | 72.85 | 33.04 | 84.27 | 74.89 | 84.91 | 92.04 | **97.05** | 95.28 | **66.47** | 81.95 | 78.28 |
| | $\mathcal{C}(\theta_v,\theta_t) + \mathcal{T}(\theta_v,\theta_t)$ | 69.02 | 34.22 | 63.29 | 73.45 | 83.34 | 88.63 | 87.23 | 93.66 | 49.45 | 71.64 | 71.39 |
| | $\mathcal{C}(\theta_v,\theta_t) - \mathcal{I}(\theta_v,\theta_t) + \mathcal{T}(\theta_v,\theta_t)$ | **73.25** | **35.1** | **86.52** | **76.24** | **86.19** | **93.12** | 95.7 | **95.58** | 65.74 | **82.41** | **78.99** |

Table 4: **Ablations on hyper-parameters:** Average top-1 accuracy across 7 datasets {DTD, EuroSAT, Food, UCF, FGVC, Caltech and Flowers} using CLIP ViT-B/16 under a 2-shot setting. Each hyperparameter is studied individually, while keeping other parameters fixed as in the main configuration.

| | 0.1 | 1 | 10 |
|---|---|---|---|
| $\lambda_{ent}$ | 77.27 | 77.29 | **77.49** |
| $\lambda_{cond}$ | 73.5 | **77.49** | 71.81 |
| $\lambda_{text}$ | **77.49** | 70.22 | 64.12 |

the marginal entropy weight $\lambda_{ent}$ consistently improves classification accuracy, reaching a maximum of 77.49% at $\lambda_{ent} = 10$. This trend suggests that stronger marginal entropy regularization—which encourages balanced class predictions across the dataset—provides beneficial guidance in the transductive setting. In the case of the conditional entropy weight $\lambda_{cond}$, performance is optimal at the intermediate value of 1, while both lower ($\lambda_{cond} = 0.1$) and higher ($\lambda_{cond} = 10$) values result in notable degradation. This finding indicates that a moderate degree of conditional entropy minimization is necessary to encourage confident predictions without leading to overfitting or instability. Lastly, the KL divergence weight $\lambda_{text}$, which controls alignment with the zero-shot prior, exhibits the highest sensitivity among the three. While small values ($\lambda_{text} = 0.1$) yield the best performance (77.49%), increasing this weight substantially reduces accuracy—falling to 70.22% at $\lambda_{text} = 1$ and to 64.12% at $\lambda_{text} = 10$. These results underscore the importance of carefully constraining the influence of the zero-shot prior to preserve the model's adaptability in low-data regimes.

Table 5: **Comparing LIMO with a TransCLIP variant that fine-tunes CLIP's text and vision encoders using the TransCLIP objective and LoRA**: Top-1 classification accuracy is reported over 11 datasets. The highest accuracy values are highlighted in **bold**, and the second highest are underlined.

| | | ImageNet | Sun | FGVC | EuroSAT | Cars | Food | Pets | Flowers | Caltech | DTD | UCF | Average |
|---|---|---|---|---|---|---|---|---|---|---|---|---|---|
| 2-shot | TransCLIP (Zanella et al., 2024) | 70.3 | 70.9 | _30.0_ | 77.1 | 71.7 | **87.0** | 91.7 | _90.6_ | 93.5 | 55.1 | _78.5_ | 74.2 |
| | TransCLIP-LoRA | **71.18** | **72.44** | 27.83 | _90.69_ | _73.33_ | 86.35 | _92.79_ | 85.22 | _94.52_ | _58.04_ | 76.28 | _75.33_ |
| | LIMO (Ours) | _71.08_ | _72.31_ | **33.8** | **91.68** | **74.47** | 86.28 | **93.78** | **92.43** | **94.94** | **63.32** | **81.21** | **77.75** |
| 4-shot | TransCLIP | 70.3 | 71.9 | _34.0_ | 79.4 | 74.0 | 86.4 | 91.6 | _93.6_ | 94.0 | _61.1_ | _79.1_ | 75.9 |
| | TransCLIP-LoRA | **71.99** | _73.8_ | 33.3 | _90.9_ | _77.3_ | **86.86** | _93.79_ | 88.47 | _95.36_ | 60.56 | 77.38 | _77.25_ |
| | LIMO (Ours) | _71.94_ | **74.08** | **40.3** | **93.1** | **78.61** | _86.59_ | **94.26** | **95.99** | **95.81** | **67.57** | **83.97** | **80.2** |
| 16-shot | TransCLIP | 71.8 | 74.7 | 38.6 | 83.0 | 79.8 | 86.9 | 92.4 | _94.4_ | 94.0 | 65.1 | 82.1 | 78.4 |
| | TransCLIP-LoRA | _73.77_ | _77.4_ | _47.98_ | _91.01_ | _86.75_ | **87.6** | _94.89_ | 93.95 | _96.66_ | _68.24_ | _83.13_ | _81.94_ |
| | LIMO (Ours) | **73.99** | **77.69** | **57.54** | **94.42** | **87.42** | _87.39_ | **95.26** | **98.73** | **97.05** | **74.73** | **89.25** | **84.86** |

**Fairly comparing LIMO with TransCLIP.** To enable a fair comparison between our transductive method, LIMO, and TransCLIP (Zanella et al., 2024), we introduce a new variant, termed TransCLIP-LoRA. In this variant, we replace the objective function of LIMO, presented in Eq. (4), with the objective function proposed in the TransCLIP paper (Zanella et al., 2024) while maintaining the same fine-tuning strategy, LoRA, as used in our approach. The TransCLIP objective function is composed of four components: (i) a cross-entropy loss applied to labeled samples, (ii) a GMM clustering loss structuring the unlabeled data, (iii) a KL divergence loss aligning the model predictions with the zero-shot predictions, and (iv) a Laplacian regularization term promoting smooth predictions across samples with similar feature representations. To balance their contributions, we assign specific weighting factors to each term; 2 to the cross-entropy loss, 2 to the GMM clustering loss, 1 to the KL divergence term, and 0.05 to the Laplacian regularization term. Experimental results for this variant are reported in Table 5.

As shown in Table 5, substituting our proposed objective function with the TransCLIP objective leads to a performance degradation of approximately 2.5% across all evaluated settings. This consistent performance drop clearly demonstrates the effectiveness of our objective function in achieving superior performances in transductive few-shot learning. Moreover, the results show that integrating the TransCLIP objective with LoRA fine-tuning strategy yields an approximate 2% improvement compared to the standard TransCLIP setting, further confirming the significant benefits of our fine-tuning strategy based on LoRA, which focuses on adapting the inner representations of the model. Overall, these findings validate the design choices underlying the proposed method LIMO—specifically, the combination of our information maximization loss function in Eq. (4) and the LoRA fine-tuning strategy—for achieving optimal performance.

## 5 Discussion

A natural question concerns our focus on the *all-to-all* setting, where training and testing share the same set of classes, and the absence of results under the *base-to-novel* setting popularized by CoOp (Zhou et al., 2022b). This choice is motivated by the fact that our proposed loss function is conceptually designed under the assumption that the support and query sets operate within the same category space. The information-maximization objective we employ promotes a balanced inter-class distribution, encouraging diversity in predictions, and preventing the model from favoring a limited subset of classes. However, as discussed in (Chiaroni et al., 2023), such a behavior is particularly effective when the class space is fixed (*all-to-all setting*) but becomes harmful on long-tailed data, where the model is required to generalize to unseen categories, as in the *base-to-new* setting. A promising direction for future work would be to extend our loss to this famous setting by redefining the mutual information term to account for unseen classes, following approaches such as in generalized category discovery Chiaroni et al. (2023), and by adjusting the text regularization term accordingly.

## 6 Conclusion

In this work, we presented a novel approach to handle transductive learning for vision-language models. We first introduced a Language-aware Information MaximizatiOn (LIMO) loss function which is based on strong text-based regularization terms. Second, we demonstrated the potential of fine-tuning a subset of the model parameters within the transductive paradigm, notably with low-rank matrices (LoRA), a perspective overlooked by current transductive methods working solely in the embedding space. We evaluated our proposed LIMO on 11 datasets and showed that it outperforms state-of-the-art transductive and inductive few-shot learning methods, highlighting new possibilities for combining PEFT and transductive learning in future research.

## Acknowledgments

This work was funded by the Natural Sciences and Engineering Research Council of Canada (NSERC). Also, we thank Calcul Québec and Compute Canada for providing compute ressources.

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

# A Ablations on Model Architecture

Table 6: **Detailed results of state-of-the-art methods in the few-shot setting for the** 11 **datasets with the ViT-B/32 as visual backbone**: Top-1 classification accuracy averaged over 3 random seeds is reported. **Bolded** values indicate highest accuracy. ✓denotes transductive methods, ✗denotes non-transductive methods.

| | | | | ImageNet | Sun | FGVC | EuroSAT | Cars | Food | Pets | Flowers | Caltech | DTD | UCF | Average |
|---|---|---|---|---|---|---|---|---|---|---|---|---|---|---|---|
| **2-shot** | Vision-only | TF (Dhillon et al., 2019) | ✓ | 34.7 | 49.5 | 19.3 | 56.5 | 37.4 | 48.7 | 47.4 | 75.1 | 83.9 | 44.5 | 57.7 | 50.4 |
| | | BD-CSPN (Liu et al., 2020) | ✓ | 39.2 | 53.1 | 20.7 | 57.2 | 42.1 | 55.5 | 55.2 | 82.4 | 86.8 | 45.6 | 61.6 | 54.5 |
| | | LaplacianShot (Ziko et al., 2020) | ✓ | 39.1 | 53.9 | 20.4 | 58.3 | 42.4 | 57.7 | 57.3 | 82.5 | 86.7 | 45.9 | 62.6 | 55.2 |
| | | PT-MAP (Hu et al., 2021b) | ✓ | 42.6 | 60.1 | 22.3 | 63.7 | 46.0 | 63.9 | 64.0 | 69.5 | 55.6 | 50.4 | 66.8 | 55.0 |
| | | TIM (Boudiaf et al., 2020b) | ✓ | 41.1 | 59.0 | 21.1 | 68.9 | 44.1 | 66.2 | 60.1 | 86.5 | 81.5 | 48.6 | 68.1 | 58.7 |
| | Vision-Language | CoOp (Zhou et al., 2022b) | ✗ | 61.3 | 63.8 | 18.6 | 62.4 | 62.3 | 73.3 | 85.8 | 82.0 | 91.2 | 49.6 | 70.4 | 65.5 |
| | | Tip-Adapter-F (Zhang et al., 2022) | ✗ | 64.9 | 66.8 | 24.6 | 67.4 | 63.7 | 80.8 | 88.2 | 85.5 | 92.8 | 54.9 | 71.8 | 69.2 |
| | | CLIP-LoRA (Zanella & Ben Ayed, 2024a) | ✗ | 65.7 | 68.6 | 24.1 | 80.8 | 64.8 | 76.3 | 86.6 | 84.7 | 93.7 | 57.4 | 75.4 | 70.7 |
| | | TransCLIP (Zanella et al., 2024) | ✓ | 64.8 | **69.5** | 22.9 | 76.9 | 63.8 | **81.2** | 89.9 | 85.4 | 92.1 | 52.9 | 71.0 | 70.0 |
| | | LIMO (Ours) | ✓ | **66** | 69.4 | **24.7** | **89.2** | **66** | 80.4 | **91.1** | **87.7** | **94.3** | **60.1** | **76.1** | **73.2** |
| **4-shot** | Vision-only | TF | ✓ | 44.5 | 59.4 | 23.2 | 62.1 | 48.6 | 60.8 | 57.9 | 85.2 | 89.1 | 52.6 | 65.2 | 59.0 |
| | | BD-CSPN | ✓ | 47.0 | 61.1 | 23.4 | 64.2 | 49.1 | 65.3 | 64.8 | 87.2 | 89.4 | 52.0 | 67.0 | 61.0 |
| | | LaplacianShot | ✓ | 46.8 | 61.1 | 23.6 | 68.4 | 49.2 | 65.6 | 66.6 | 87.6 | 89.3 | 51.4 | 67.5 | 61.6 |
| | | PT-MAP | ✓ | 50.1 | 65.5 | 24.1 | 68.9 | 52.3 | 70.3 | 69.0 | 73.3 | 57.3 | 56.1 | 70.1 | 59.7 |
| | | TIM | ✓ | 50.4 | 65.0 | 24.7 | 70.0 | 56.1 | 73.0 | 74.4 | 90.5 | 88.7 | 55.9 | 71.8 | 65.5 |
| | Vision-Language | CoOp | ✗ | 63.2 | 67.1 | 24.0 | 68.7 | 66.2 | 75.6 | 88.8 | 87.9 | 93.0 | 55.3 | 75.0 | 69.5 |
| | | Tip-Adapter-F | ✗ | 65.8 | 68.3 | 28.8 | 71.5 | 67.6 | 80.9 | 88.6 | 88.9 | 94.6 | 58.0 | 75.1 | 71.6 |
| | | CLIP-LoRA | ✗ | 66.5 | 70.3 | 27.7 | 85.6 | 68.3 | 75.6 | 86.3 | 90.1 | 94.3 | 60.3 | 76.5 | 72.9 |
| | | TransCLIP | ✓ | 64.7 | 70.1 | 26.4 | 78.0 | 66.5 | 80.3 | 87.2 | 88.7 | 92.2 | 58.0 | 74.3 | 71.5 |
| | | LIMO (Ours) | ✓ | **66.9** | **71.6** | **29** | **90.8** | **70.8** | 80.8 | **91.4** | **93.4** | **95.2** | **63.5** | **79.3** | **75.7** |
| **16-shot** | Vision-only | TF | ✓ | 55.6 | 68.0 | 29.7 | 69.7 | 62.9 | 72.6 | 73.7 | 92.0 | 91.6 | 61.6 | 73.1 | 68.2 |
| | | BD-CSPN | ✓ | 55.3 | 67.5 | 29.8 | 69.5 | 62.3 | 72.9 | 74.2 | 91.9 | 91.7 | 59.6 | 73.3 | 68.0 |
| | | LaplacianShot | ✓ | 54.8 | 66.7 | 28.4 | 71.2 | 60.9 | 73.2 | 75.3 | 91.3 | 91.3 | 58.3 | 72.9 | 67.7 |
| | | PT-MAP | ✓ | 56.9 | 69.9 | 29.2 | 71.3 | 63.1 | 74.1 | 78.7 | 77.1 | 60.7 | 61.9 | 72.9 | 65.1 |
| | | TIM | ✓ | 60.5 | 71.8 | 33.0 | 79.4 | 72.2 | 78.1 | 85.0 | 92.8 | 88.4 | 66.6 | 78.1 | 73.3 |
| | Vision-Language | CoOp | ✗ | 66.8 | 72.2 | 32.9 | 83.3 | 76.0 | 78.6 | 88.7 | 95.4 | 94.9 | 65.3 | 78.6 | 75.7 |
| | | Tip-Adapter-F | ✗ | 68.4 | 74.1 | 34.8 | 83.4 | 77.0 | 81.7 | 90.4 | 94.3 | 95.1 | 68.0 | 80.5 | 77.1 |
| | | CLIP-LoRA | ✗ | 68.4 | 74.0 | 44.9 | 91.8 | 79.7 | 78.2 | 88.8 | 96.2 | 95.2 | 68.2 | 82.8 | 78.9 |
| | | TransCLIP | ✓ | 66.6 | 72.6 | 30.1 | 78.9 | 73.2 | 81.1 | 89.5 | 90.9 | 94.4 | 62.7 | 77.2 | 74.3 |
| | | LIMO (Ours) | ✓ | **69** | **75.3** | **45.8** | **93.6** | **81.9** | **81.9** | **92.5** | **97.4** | **96.3** | **72.5** | **85.8** | **81.1** |

Table 7: **Detailed results of state-of-the-art methods in the few-shot setting for the** 11 **datasets with the ViT-L/14 as visual backbone**: Top-1 classification accuracy averaged over 3 random seeds is reported. **Bolded** values indicate highest accuracy. ✓denotes transductive methods, ✗denotes non-transductive methods.

| | | | | ImageNet | Sun | FGVC | EuroSAT | Cars | Food | Pets | Flowers | Caltech | DTD | UCF | Average |
|---|---|---|---|---|---|---|---|---|---|---|---|---|---|---|---|
| **2-shot** | Vision-only | TF (Dhillon et al., 2019) | ✓ | 50.1 | 56.6 | 33.5 | 71.7 | 58.3 | 71.6 | 65.7 | 93.0 | 90.5 | 49.8 | 69.4 | 64.6 |
| | | BD-CSPN (Liu et al., 2020) | ✓ | 57.0 | 61.2 | 35.6 | 72.6 | 65.1 | 79.9 | 77.2 | 95.7 | 92.8 | 52.3 | 74.7 | 69.5 |
| | | LaplacianShot (Ziko et al., 2020) | ✓ | 56.5 | 61.3 | 35.9 | 76.8 | 65.4 | 80.3 | 77.4 | 96.2 | 93.3 | 52.4 | 74.8 | 70.0 |
| | | PT-MAP (Hu et al., 2021b) | ✓ | 61.3 | 68.0 | 37.0 | 78.4 | 68.4 | 87.3 | 86.7 | 77.9 | 61.1 | 56.5 | 75.2 | 68.9 |
| | | TIM (Boudiaf et al., 2020b) | ✓ | 59.7 | 67.6 | 35.4 | 82.2 | 69.3 | 87.4 | 85.5 | 95.1 | 91.4 | 53.2 | 78.6 | 73.2 |
| | Vision-Language | CoOp (Zhou et al., 2022b) | ✗ | 73.1 | 69.3 | 38.8 | 72.7 | 81.2 | 89.4 | 93.5 | 93.5 | 95.0 | 59.2 | 80.0 | 76.9 |
| | | Tip-Adapter-F (Zhang et al., 2022) | ✗ | 76.6 | 72.6 | 42.2 | 76.7 | 80.4 | 91.1 | 93.8 | 93.8 | 95.5 | 60.3 | 80.4 | 78.5 |
| | | CLIP-LoRA (Zanella & Ben Ayed, 2024a) | ✗ | 77.3 | 75.5 | 42.4 | 85.9 | 82.3 | 89.8 | 93.2 | 94.7 | 96.8 | 67.3 | 84.2 | 80.9 |
| | | TransCLIP (Zanella et al., 2024) | ✓ | 76.8 | 75.1 | 40.0 | 82.1 | 79.9 | **91.8** | 95.0 | **96.6** | 95.9 | 62.6 | 83.2 | 79.9 |
| | | LIMO (Ours) | ✓ | **77.8** | **76.5** | **46.1** | **93.4** | **83.6** | 91.5 | **95.5** | 96.5 | **97.1** | **71.4** | **85.3** | **83.2** |
| **4-shot** | Vision-only | TF | ✓ | 61.6 | 66.5 | 40.6 | 71.4 | 69.6 | 81.9 | 79.0 | 96.4 | 94.4 | 58.5 | 77.5 | 72.5 |
| | | BD-CSPN | ✓ | 64.3 | 67.8 | 40.6 | 71.4 | 72.2 | 84.7 | 82.8 | 96.7 | 95.2 | 56.9 | 79.6 | 73.8 |
| | | LaplacianShot | ✓ | 63.8 | 67.6 | 40.0 | 78.9 | 72.0 | 85.4 | 85.7 | 97.3 | 95.2 | 56.7 | 79.6 | 74.7 |
| | | PT-MAP | ✓ | 68.0 | 72.7 | 41.7 | 77.4 | 73.8 | 88.9 | 89.9 | 78.3 | 62.9 | 60.1 | 79.2 | 72.1 |
| | | TIM | ✓ | 68.9 | 72.7 | 42.0 | 78.4 | 77.8 | 90.0 | 92.3 | 97.4 | 91.1 | 63.5 | 83.7 | 78.0 |
| | Vision-Language | CoOp | ✗ | 74.9 | 73.1 | 43.6 | 75.9 | 83.3 | 88.7 | 94.6 | 95.9 | 96.5 | 63.9 | 82.8 | 79.4 |
| | | Tip-Adapter-F | ✗ | 77.1 | 74.1 | 47.4 | 81.4 | 82.3 | 91.2 | 94.0 | 95.5 | 96.5 | 64.4 | 83.9 | 80.7 |
| | | CLIP-LoRA | ✗ | 77.9 | 76.7 | 48.9 | 86.4 | 85.2 | 89.6 | 93.9 | 97.4 | 97.2 | 70.4 | 86.4 | 82.7 |
| | | TransCLIP | ✓ | 76.9 | 76.2 | 45.9 | 81.5 | 81.2 | 91.4 | 94.3 | 98.2 | 96.1 | 66.8 | 84.9 | 81.2 |
| | | LIMO (Ours) | ✓ | **78.6** | **78** | **53.1** | **94.2** | **86.6** | **91.9** | **95.8** | **98.2** | **97.6** | **74.1** | **88.3** | **85.1** |
| **16-shot** | Vision-only | TF | ✓ | 71.1 | 74.9 | 50.1 | 78.6 | 81.5 | 88.1 | 88.6 | 98.5 | 96.1 | 67.3 | 83.0 | 79.8 |
| | | BD-CSPN | ✓ | 71.1 | 74.4 | 49.4 | 78.1 | 81.2 | 88.0 | 89.8 | 98.4 | 95.8 | 66.5 | 82.5 | 79.6 |
| | | LaplacianShot | ✓ | 69.8 | 72.7 | 47.0 | 81.7 | 80.2 | 88.0 | 90.1 | 98.0 | 95.7 | 63.3 | 82.8 | 79.0 |
| | | PT-MAP | ✓ | 72.9 | 75.9 | 48.1 | 79.1 | 79.9 | 89.4 | 92.0 | 80.5 | 66.0 | 65.6 | 80.5 | 75.4 |
| | | TIM | ✓ | 76.4 | 78.7 | 52.5 | 89.4 | 86.5 | 91.0 | 92.0 | 98.2 | 94.5 | 73.2 | 84.8 | 83.4 |
| | Vision-Language | CoOp | ✗ | 78.2 | 77.5 | 55.2 | 88.3 | 89.0 | 89.8 | 94.6 | 99.1 | 97.2 | 74.4 | 87.3 | 84.6 |
| | | Tip-Adapter-F | ✗ | 79.3 | 79.6 | 55.8 | 86.1 | 88.1 | 91.6 | 94.6 | 98.3 | 97.5 | 74.0 | 87.4 | 84.8 |
| | | CLIP-LoRA | ✗ | 79.6 | 79.4 | 66.2 | 93.1 | 90.9 | 89.9 | 94.3 | 99.0 | 97.3 | 76.5 | 89.9 | 86.9 |
| | | TransCLIP | ✓ | 77.8 | 78.7 | 53.0 | 84.4 | 86.3 | 91.6 | 94.8 | 98.8 | 97.3 | 71.2 | 86.5 | 83.7 |
| | | LIMO (Ours) | ✓ | **80.4** | **81.1** | **69** | **95** | **92** | **92** | **96.2** | **99.2** | **98.1** | **79.6** | **91.8** | **88.6** |

