# OpenReview forum: "Language-Aware Information Maximization for Transductive Few-Shot CLIP"
_TMLR — Accepted by TMLR_

### Review · Reviewer_a4bx · 2025-10-02

**Summary Of Contributions:**

Contribution summary:

- Novel loss function: The paper proposes a language-aware information maximization objective with three terms:

Mutual information between vision inputs and textual class descriptions (conditional + marginal entropy)
KL divergence regularization that penalizes deviation from zero-shot predictions
Standard cross-entropy on labeled support samples

- PEFT exploration: The work challenges conventional practices by applying Low-Rank Adaptation (LoRA) to fine-tune internal model parameters in the transductive setting. Prior transductive VLM methods only operated on the output embedding space.

- Strong empirical results: LIMO achieves consistent improvements across 11 datasets, outperforming TransCLIP by ~5% on average in 4-shot and showing gains over inductive methods like CLIP-LoRA.

Key strengths:
- The method provides a principled generalization of information maximization to VLMs by incorporating textual modality.

- Comprehensive ablations demonstrate each component's contribution.

-Results hold across different vision encoders (ViT-B/16, ViT-B/32, ViT-L/14).

-The approach is orthogonal and can enhance other methods (CoOp, ProLIP).

- Fixed hyperparameters across all experiments suggest good generalizability.

Key weaknesses:
- Improvements are sometimes modest (2-3% in certain settings), raising questions about practical significance.

- Limited analysis of why PEFT helps in transductive settings beyond empirical demonstration.

- No investigation of which dataset characteristics or task properties lead to larger gains.

- Sensitivity analysis (Table 4) shows high sensitivity to λ_text but is only conducted on one dataset.

- Computational costs are mentioned briefly (5 minutes added) but not thoroughly analyzed across different scales.

- The comparison with TransCLIP-LoRA (Table 5) somewhat conflates the contributions of the loss function versus the fine-tuning strategy.

**Audience:**

Yes

**Audience Explanation:**

Yes, the TMLR audience would likely be interested in these findings, as the paper addresses the timely intersection of vision-language models, few-shot learning, and parameter-efficient fine-tuning—all active research areas—and demonstrates that applying PEFT strategies like LoRA to transductive settings yields consistent empirical improvements, challenging the conventional practice of only fine-tuning output embeddings in transductive few-shot learning.

**Claims And Evidence:**

Yes

**Claims Explanation:**

The claims are **generally supported but with some limitations**. The core performance claims are backed by comprehensive experiments across 11 datasets with three shot settings (2/4/16-shot) and multiple backbones (Tables 1, 2, 6, 7), showing consistent improvements over baselines. The ablation studies (Table 3) provide reasonable evidence that each loss component contributes to performance. However, several gaps weaken the evidence: (1) **no statistical significance testing** despite results averaged over only 3 seeds and some improvements being modest (1-2%); (2) **limited hyperparameter analysis**, with sensitivity tests (Table 4) conducted only on one dataset (DTD) despite claiming fixed hyperparameters generalize across all tasks; (3) **unclear attribution** in the TransCLIP-LoRA comparison (Table 5), which conflates the contributions of the loss function versus the fine-tuning strategy; and (4) **lack of analysis** explaining *why* PEFT helps in transductive settings or *which* dataset characteristics lead to larger gains. The evidence is convincing enough to support that LIMO improves performance in most settings, but the depth of understanding about when, why, and by how much (with statistical confidence) remains somewhat superficial.

**Requested Changes:**

- Add error bars to all tables and figures, and provide statistical tests (e.g., paired t-tests) to demonstrate that improvements over baselines are statistically significant, especially given only 3 random seeds and some modest gains (1-3%).

- The claim that λ_ent=10, λ_cond=1, λ_text=0.1 generalize across all datasets is only validated on DTD (Table 4). Test sensitivity on at least 3-5 diverse datasets to substantiate this claim, or acknowledge dataset-specific tuning may be needed.

- In Table 5, the TransCLIP-LoRA comparison conflates two factors (loss function vs. fine-tuning strategy). Add an ablation showing: (a) TransCLIP loss + output layer only, (b) TransCLIP loss + LoRA, (c) LIMO loss + output layer only, (d) LIMO loss + LoRA, to clearly attribute gains.

- Investigate which dataset characteristics (domain shift, fine-grainedness, number of classes) correlate with larger LIMO gains to provide practical guidance.

- Discuss when LIMO underperforms (e.g., StanfordCars in 2-shot Table 1) and provide hypotheses about why.

---

> ### Author Response · Authors · 2025-11-29
>
> We sincerely appreciate the thoughtful  feedback of the reviewer, and address the raised concerns below.
>
> Sensitivity analysis of the hyperparameters: We performed additional experiments on six diverse datasets (EuroSAT, Food101, UCF101, FGVC-Aircraft, Caltech101 and OxfordFlowers10) to assess the generalization of the proposed configuration across datasets. The obtained results show trends in line with those observed with DTD, reinforcing our original findings. These results have been included in the revised version in Table 4 and further discussed in the paragraph “Sensitivity Analysis of LIMO’s Hyperparameters.”, where we detail the influence of each regularization weight.
>
> On the comparison with Transclip in Table 5:  In fact, we mentioned in the introduction that prior transductive few-shot CLIP methods, including TransCLIP, operate on the output embedding space, following a common practice in the abundant vision-only few-shot literature (i.e., only the final, linear-classifier layer is updated while the remaining model parameters are frozen). In contrast, our objective in this paper was to design a loss function that can effectively leverage the inner representations of the network through a parameter efficient fine tuning method (LoRA), thereby challenging this convention. Accordingly, Table 5 compares the proposed loss function and the one of TranCLIP under the same LoRA-based fine tuning strategy. Therefore, the TransCLIP variant we evaluated in Table 5 isolates the effect of the loss function and is, indeed, purposely made different from the linear-classifier variant in the original TransCLIP paper. The results show that LIMO’s objective function better supports inner-representations fine-tuning, consistently yielding higher performance. Moreover, as reported in Table 3, LIMO remains effective under other fine-tuning strategies, confirming its orthogonality and broadly applicable design.
>
> On the correlation between LIMO’s gains and dataset characteristics:
> The revised version now includes zero-shot results. From these results  we can clearly see that datasets exhibiting a large distributional shift from CLIP’s pretraining distribution such as Eurosat, DTD and FGVC show the largest improvement compared to other datasets. This is consistent with LIMO’s objective which maximizes mutual information between visual inputs and textual class representations, particularly effective when the default alignment between CLIP’s encoders is weak (the case for those datasets).
> In regard to the Food101 dataset (and not the StanfordCars) in the 2-shot setting: The zero-shot performance indicates that CLIP already performs strongly on this dataset. Under such conditions, the scarce data provided by the support set, with only two samples per class, may result in over-regularization by LIMO. This may explain the slight drop in LIMO’s performance for this specific dataset, as observed in Table 1.  We added a paragraph explaining LIMO’s behaviour in section 4.3 in the revised paper.
>
> Addressing the error bars: As suggested by the reviewer, we conducted additional experiments and included a new plot in the revised version (section 4.4). This figure reports conventional 95% confidence intervals, averaged over four representative datasets including fine-grained (FGVC), texture (DTD), satellite (Eurosat) and generic (Caltech) across 10 seeds and for the three shot settings considered in this work (2, 4 and 16 shots). The results compare the top three methods: TransCLIP, CLIP-LoRA, and our proposed LIMO. In this plot, one may observe that there is no overlap between our method and the two other methods, indicating that LIMO’s improvements are statistically significant relative to the state of the art.

---

### Review · Reviewer_Naxi · 2025-10-04

**Summary Of Contributions:**

The paper proposes a fine-tuning loss function for CLIP methods, given a labeled dataset of image-prompt pairs, and an unlabeled dataset of images.The work tackles the transductive setting, under which the structure of the whole unlabel dataset is leveraged when fine-tuning, in opposition to the inductive setting, where samples are considered separately. LIMO, the loss function that they propose has three terms: the cross-entropy term, which is computed on the labeled dataset; the mutual information term, which is computed on the unlabeled dataset; and the text-based regularization term, which is proportional to the sum of KL divergences between the distribution of the model and the base distribution.

Transductive methods are common in vision-only settings, but not as much in vision-language settings (with a single prior work that the authors compare to). In the experiments of the paper, LIMO outperforms all the baselines in most settings.

**Strengths**:
The method introduced improves over the baselines on a suite of 12 datasets. The experiments of the paper are very complete: the authors ablate with respect to having only some of the three terms in the loss, with respect to three fine-tuning strategies, and with respect to different values of the multipliers of each loss, and with respect to the number of samples per class.

**Weaknesses**:
There are no apparent weaknesses. While not highly novel technically, this is a very strong submission that shows state-of-the-art fine-tuning results.

**Audience:**

Yes

**Audience Explanation:**

Anyone that wants to fine-tune a CLIP model with labeled and unlabeled data may be interested in the paper.

**Broader Impact Concerns:**

None.

**Claims And Evidence:**

Yes

**Claims Explanation:**

The writing is clear, and the experiments are very complete.

**Requested Changes:**

- Use \citep when needed.

- In Table 1 and throughout the text, the authors compare to vision-only fine-tuning methods. Could you describe the vision-only fine-tuning methods more precisely? In particular, in Table 1, does the term “vision-only” refer to the fact that only the visual embedding is trained?

---

> ### Author Response · Authors · 2025-11-29
> **Clarification on Vision-Only Terminology and Citation Style Corrections**
>
> We thank the reviewer for acknowledging the strengths of our paper.
> Regarding the requested clarification, the term vision-only refers to methods that employ only a vision encoder, in contrast to vision-language models that additionally use a text encoder. In these methods, fine tuning is applied exclusively to visual embeddings as no textual embeddings are involved.
> Concerning the citation formatting issue, we appreciate the reviewer’s observation and, as suggested, have replaced all occurrences of \cite with \citep in the revised version to ensure that all citations are properly enclosed in brackets.

---

### Review · Reviewer_nHa9 · 2025-11-04

**Summary Of Contributions:**

Summary:

- This paper proposes LIMO, a transductive few-shot learning method for CLIP, which introduces a novel loss function that calculates mutual information between visual inputs and textual class representations. Comprehensive evaluations across 11 datasets demonstrate that LIMO outperforms state-of-the-art methods by significant margins.

Strengths:

- It integrates mutual information with text-based regularization, creating a principled objective function for transductive settings.
- Comprehensive experiments across multiple datasets and shot settings demonstrate the superiority of the proposed method.
- Ablation studies highlight the importance of each loss component.

Weaknesses:
- The fundamental concept of the mutual information objective is not new. This loss term appears to be a direct adaptation of a prior vision-only method [1], primarily by substituting the predictive logits with vision-language ones from CLIP.
- The evaluation is conducted exclusively in an *all-to-all* setting, where training and testing occur on the same set of classes. This fails to assess the method's performance in the widely used *base-to-novel* setting (as established in CoOp).

[1] Transductive Information Maximization For Few-Shot Learning, NeurIPS 2020

**Audience:**

Yes

**Audience Explanation:**

Yes, the integration of mutual information with text-based regularization presents a practical and valuable approach for adapting vision-language models in transductive learning settings.

**Broader Impact Concerns:**

This work presents a fine-tuning method for vision-language models with few immediate ethical concerns.

**Claims And Evidence:**

Yes

**Claims Explanation:**

The claims in the submission are well-supported by comprehensive evidence from extensive experiments across 11 datasets and multiple few-shot settings.

**Requested Changes:**

- It is encouraged to include some recent methods such as [2, 3].
- Experiments on the widely used *base-to-novel* setting [3] would strengthen the evaluation.
- There are some writing issues:
   - Add brackets for citations;
   - Replace `'prompts'' with "prompts"  on Page 5;
   - Resolve the inconsistent notation symbols between Eq. (4) and Tab. 3;
   - Verify and correct references (e.g., "Information maximization for few-shot learning" should be "Transductive information xx" on Page 12).

[2] MMA: Multi-Modal Adapter for Vision-Language Models, CVPR 2024

[3] Rethinking Few-Shot Adaptation of Vision-Language Models in Two Stages, CVPR 2025

---

> ### Author Response · Authors · 2025-11-29
> **Clarifications on Mutual-Information Novelty, Base-to-Novel Evaluation, and Writing Corrections**
>
> We highly value the thoughtful  comments of the reviewer, and provide detailed responses to the raised concerns below.
>
> On the technical novelty, we concede that the fundamental concept of mutual information is not new, and our formulation builds upon prior vision-only work. However, our contribution lies in extending this concept to the vision-language setting by: i) adding to the mutual information a Kullback-Leibler (KL) divergence penalizing deviation of the network’s probabilistic outputs from the text-driven zero-shot predictions; and ii) reformulating the objective function to maximize the mutual information between the vision inputs and the textual class descriptions. Despite its simplicity, this formulation is still new and demonstrates consistent effectiveness.
>
> Discussing the base-to-novel setting, we appreciate the reviewer’s suggestion to include evaluations under the base-to-novel setting (as established in CoOp). We added discussions on this in the conclusion section. Indeed, our proposed Language-aware Information MaximizatiOn (LIMO) loss function is conceptually designed for the all-to-all setting, where both support and query samples share the same category space (Y_s = Y_q with Y denoting the set of classes). In fact, the information-maximization objective we employ  promotes a balanced inter-class distribution, encouraging diversity in predictions, and preventing the model from favoring a limited subset of classes. However, as discussed in [1], such a behavior is particularly effective when the class space is fixed (all-to-all setting) but becomes harmful on long-tailed data, where the model is required to generalize to unseen categories,  as in the base-to-new setting. An interesting future direction would be to extend our loss function to this  setting by redefining the mutual information term to handle unseen classes like in [1] and by adjusting the text regularization term accordingly.
>
> Addressing writing issues, we thank the reviewer for pointing to these issues. In the revised version we have added brackets for all citations, replaced `'prompts'' with "prompts" on Page 5 and resolved the inconsistent notations involving Eq. (4) and Tab. 3. We also rectified the reference mentioned by the reviewers (thanks!).
>
> [1] Chiaroni, Florent, et al. "Parametric information maximization for generalized category discovery." Proceedings of the IEEE/CVF international conference on computer vision. 2023.

---

### Decision · Action_Editor_pntT · 2026-02-07

**Recommendation:** Accept as is

**Audience:**

Yes

**Audience Explanation:**

The paper addresses a timely and underexplored problem at the intersection of transductive learning, few-shot adaptation, and vision–language models. Researchers and practitioners working on CLIP, VLM adaptation, semi-supervised or transductive learning, and parameter-efficient fine-tuning would find the findings valuable, particularly the demonstrated benefits of information-theoretic objectives and PEFT strategies in transductive VLM settings.

**Claims And Evidence:**

Yes

**Claims Explanation:**

Reviewers generally agreed that the empirical evidence is strong and clearly supports the paper’s claims. The evaluation spans a large number of datasets, shot settings, and model backbones, and is complemented by ablation studies, sensitivity analyses, and additional statistical reporting in the revised version. While one reviewer expressed concerns that the technical contribution is incremental in nature, this concern was explicitly about novelty rather than evidential support. In general, the experimental results are sound and that the proposed method consistently improves performance, indicating that the claims themselves are well supported by the evidence.